# A Hydrofluoric Acid-Free Green Synthesis of Magnetic M.Ti_2_CT_x_ Nanostructures for the Sequestration of Cesium and Strontium Radionuclide

**DOI:** 10.3390/nano12183253

**Published:** 2022-09-19

**Authors:** Jibran Iqbal, Kashif Rasool, Fares Howari, Yousef Nazzal, Tapati Sarkar, Asif Shahzad

**Affiliations:** 1College of Natural and Health Sciences, Zayed University, Abu Dhabi 144534, United Arab Emirates; 2Qatar Environment and Energy Research Institute, Hamad Bin Khalifa University (HBKU), Qatar Foundation, Doha P.O. Box 5824, Qatar; 3Department of Materials Science and Engineering, Uppsala University, Box 35, SE-75103 Uppsala, Sweden

**Keywords:** MAX phase, magnetic nanostructure, radionuclide, alkalization, radioactive waste, water treatment

## Abstract

**Highlights:**

**Abstract:**

MAX phases are the parent materials used for the formation of MXenes, and are generally obtained by etching using the highly corrosive acid HF. To develop a more environmentally friendly approach for the synthesis of MXenes, in this work, titanium aluminum carbide MAX phase (Ti_2_AlC) was fabricated and etched using NaOH. Further, magnetic properties were induced during the etching process in a single-step etching process that led to the formation of a magnetic composite. By carefully controlling etching conditions such as etching agent concentration and time, different structures could be produced (denoted as *M.Ti_2_CT_x_*). Magnetic nanostructures with unique physico-chemical characteristics, including a large number of binding sites, were utilized to adsorb radionuclide Sr^2+^ and Cs^+^ cations from different matrices, including deionized, tap, and seawater. The produced adsorbents were analyzed using X-ray diffraction (XRD), scanning electron microscopy (SEM), X-ray energy dispersive spectroscopy (EDS), and X-ray photoelectron spectroscopy (XPS). The synthesized materials were found to be very stable in the aqueous phase, compared with corrosive acid-etched MXenes, acquiring a distinctive structure with oxygen-containing functional moieties. Sr^2+^ and Cs^+^ removal efficiencies of *M.Ti_2_CT_x_* were assessed via conventional batch adsorption experiments. *M.Ti_2_CT_x_-A_III_* showed the highest adsorption performance among other *M.Ti_2_CT_x_* phases, with maximum adsorption capacities of 376.05 and 142.88 mg/g for Sr^2+^ and Cs^+^, respectively, which are among the highest adsorption capacities reported for comparable adsorbents such as graphene oxide and MXenes. Moreover, in seawater, the removal efficiencies for Sr^2+^ and Cs^+^ were greater than 93% and 31%, respectively. Analysis of the removal mechanism validates the electrostatic interactions between *M.Ti_2_C-A_III_* and radionuclides.

## 1. Introduction

Nuclear energy is a prevalent, sustainable, cost-effective, and green source of energy [1]. Nevertheless, the environmental effect of producing nuclear power through nuclear fission reactions is an unsettled matter. Spent nuclear fuel that consists of a large amount of waste full of toxic radioisotopes poses a huge environmental challenge [2]. Nuclear catastrophes including Chernobyl and Fukushima have caused considerable harm to water bodies [2]. Radioactive strontium (two isotopes: ^90^Sr and ^89^Sr) and cesium (^137^Cs) are the most predominant nuclear fission products [3], generated by the nuclear fission of larger isotopes including uranium (^235^U) [3,4]. Radioactive ^90^Sr and ^137^Cs are major contributors to radiotoxicity in nuclear waste that could be discharged into water bodies. The isotopes ^90^Sr and ^137^Cs have half-lives of approximately 29 and 30.17 years, and emit strong beta- and gamma-rays, respectively. In addition to the emission of these extremely harmful rays, high solubility and chemical reactivity of both ^90^Sr and ^137^Cs could cause accidental release, and consequently, their occurrence in seawater and/or wastewater is a subject of great concern and needs to be tackled immediately and effectively [5,6].

Several demonstrative methodologies have been adopted over previous decades to clean radioactive ^90^Sr- and ^137^Cs-contaminated wastewater including solvent extraction, chemical precipitation, electrodialysis, membrane filtration, coagulation, and adsorption [7,8,9]. Nevertheless, the adsorption process has dominance over other approaches for targeted adsorption in water bodies since it is harmless with no secondary pollution. Furthermore, the adsorption process is especially suited for the removal of trace radionuclides in the presence of other ions such as Na^+^, K^+^, Ca^2+^, and Mg^2+^ [10]. In the adsorption process, the most important factor is the design and synthesis of appropriate adsorbents that are economical, chemically stable, more selective, and easier to prepare.

In recent years, the application of different engineered micro/nanomaterials in the adsorption of Cs and Sr has been extensively studied [11,12,13]. In particular, two-dimensional (2D) nanomaterials, such as graphene oxide (GO) and layered double hydroxides (LDHs) [14], are considered effective adsorbents for radionuclides Sr and Cs, due to their outstanding adsorption efficiencies. Nevertheless, there are continuous attempts to synthesize better materials with very high adsorption capacities, higher selectivity, and the ability to adsorb radionuclides from different aqueous matrices. However, these materials, including titanosilicate zeolites, functionalized silica monoliths, mesoporous silica, organic ligands, and their nanocomposites/derivatives, are expensive or ineffective in treating large amounts of nuclear liquid waste, especially in the presence of high concentrations of co-existing cations [15,16,17,18,19].

Since 2011, a new family of 2D materials, generally known as MXenes and comprising of transition metal carbides, nitrides, and carbonitrides, are being synthesized, and these are analogous to GO nanosheets [20,21]. These are described by the general formula *M_n+1_X_n_T_x_* and fabricated by *A* layer etching from MAX phases (*M_n+1_AX_n_*, where *M* represents early transition metals, *A* represents the group IIIA elements, X could be carbon, nitrogen, or both in the periodic table of elements, and *n* could be 1, 2, 3, or 4) [22,23]. These 2D-layered structures have remarkable physico-chemical and mechanical characteristics and demonstrate hydrophilic behavior and oxygenated terminal groups that are immediately available as sorption sites [24,25,26]. Consequently, such intriguing features make them ideal adsorbents for radionuclide-contaminated water remediation [25,27]. Many different MXenes have been prepared and utilized for different applications. However, the production of MXenes from MAX phases requires an etchant such as HF to remove the *A* layer from layered *M_n+1_AX_n_* geometry to produce 2D *M_n+1_X_n_* (MXene). Consequently, nearly all MXenes, including titanium carbide Ti_2_C (2:1 phase), are produced using toxic and dangerous HF [28,29,30]. HF is not an environmental-friendly etchant and causes severe environmental issues [31]. Therefore, it is necessary to replace HF with a more environmentally friendly approach.

Magnetic materials such as Fe_2_O_3_ and Fe_3_O_4_ nanoparticles have potential applications in water treatment [32]. Strongly magnetic materials can offer an easy and efficient approach for their separation after the adsorption of contaminants [33,34]. After application, the magnetic adsorbent can be recovered and reused in industrial processes. Therefore, if the exfoliated MAX phases could be made magnetic, this will facilitate separation and regeneration with a complimentary increase in the stability of the materials. In this work, we have fabricated nanostructured materials by exfoliating Ti_2_AlC (211) using hydrothermal treatment, and the magnetic property was induced during the exfoliation process due to the formation of a Fe_3_O_4_-based composite. The produced magnetic materials were characterized and utilized for radionuclide Sr^2+^ and Cs^+^ removal, and the adsorption efficiencies in different matrices including seawater were evaluated. 

## 2. Materials and Methods

### 2.1. Synthesis of M.Ti_2_CT_x_

To synthesize the magnetic *M.Ti*_2_*CT_x_* nanostructures, 0.25 g FeSO_4_·7H_2_O was dissolved in 40 mL deionized water in a 100 mL beaker. Afterward, a certain amount of NaOH was inserted into the solution, and the solution was stirred to produce a homogenous suspension. A 0.2 g measure of Ti_2_AlC MAX phase was then added to the solution and reacted for 1 h at room temperature. The synthesis method of the Ti_2_AlC MAX phase is reported in our previous work [35]. The prepared suspension was filled into a Teflon-lined stainless autoclave and treated at 200 °C in an oven. After 12 h of hydrothermal treatment, the prepared material was washed repeatedly with DI water and ethanol. The obtained blackish/grayish residues were collected and dried at 60 °C overnight in a vacuum oven. The synthesis protocol was varied to optimize the process and achieve materials with high adsorption capacities for radionuclides. 

As a reference material, Fe_3_O_4_ magnetic particles and Alk-Ti_2_C_sheet_ were also synthesized using the same procedure used for *M.Ti_2_CT_x_-A_III_*. However, for Fe_3_O_4_ synthesis the Ti_2_AlC MAX phase was not added during synthesis. For Alk-Ti_2_C_sheet_ fabrication, the Ti_2_AlC MAX phase was exfoliated in 5 M NaOH at 200 °C for 12 h in the absence of FeSO_4_.7H_2_O.

### 2.2. Characterization

The surface morphology and structure of the Ti_2_AlC MAX phases and as-synthesized *M.Ti_2_C-A_III_* powders were analyzed using a field emission scanning electron microscope (SEM, S-4800, HITACHI, Tokyo, Japan). The samples were gold-coated with a Balzers’ sputtering device prior to analysis with SEM. The X-ray powder diffraction spectra of the synthesized materials were recorded using Rigaku D/MAX 2500PC powder XRD (Rigaku, Tokyo, Japan) in a scan range of 2–80°. The accelerating voltage and current were set at 40 kV and 200 mA, respectively, with a monochromatic Cu Kα radiation of wavelength (λ = 1.5405 Å). A superconducting quantum interference device magnetometer (Quantum Design, San Diego, CA, USA) was used for the magnetic characterization of Fe_3_O_4_ and the final nanostructure. Inductively coupled plasma optical emission/mass spectrometry (ICP-MS, Perkin Elmer, Waltham, MA, USA) was used to analyze the strontium and cesium ion concentrations in the solutions. The surface area and pore size analyses were conducted using a Brunauer–Emmett–Teller (BET) analyzer and Barrett−Joyner−Halen (BJH) method, respectively. The BJH method was applied using a Micromeritics ASAP-2020 analyzer with a nitrogen gas adsorption–desorption isotherm at 77 K to determine the pore size distribution. XPS spectra of as-prepared magnetic composites were measured using a scanning X-ray micrograph (SXM: ULVAC-PHI II, Quantera, Kanagawa, Japan). For XPS spectra after Cs and Sr adsorption, the sample was prepared by inserting 10 mg of as-prepared *M.Ti_2_C-A_III_* into a binary solution containing 5 ppm of both Sr^2+^ and Cs^+^. After a 12 h reaction, the adsorbent was separated, washed, and dried in an oven at 50 °C under vacuum conditions.

### 2.3. Adsorption Experiments

To evaluate the radionuclides removal capabilities of the synthesized structures, adsorption testing was performed in batch experiments with specific pH and adsorbent amounts. The adsorbent was then removed and filtered, and the remaining concentrations of Sr^2+^ and Cs^+^ were estimated using inductively coupled plasma optical emission spectroscopy (ICP-OES, Perkin Elmer, Waltham, MA, USA) and ICP-MS, as appropriate. The absolute adsorption capacity and adsorption efficiency of the cations were calculated using the following Equations (1) and (2):(1)$$Q_{e}=\frac{\left(C_{o}-C_{e}\right)\times V}{m}$$
(2)$$Q(\%)=\frac{\left(C_{o}-C_{e}\right)}{V}\times 100$$
where *C_o_* is the initial concentration in ppm, and *C_e_* is the final concentration of Sr^2+^ and Cs^+^, respectively; *V* is the volume of solution in liter, *m* is the mass of the adsorbent (g); and *Q_e_* is the adsorption capacity of the cations.

Experiments were performed to assess the influence of pH on both Cs^+^ and Sr^2+^ adsorption in a pH range 2.0–9.0. Batch experiments were carried out using different concentrations (10–1000 mg/L) at room temperature and pH of 6 for 6 h of contact time. Pseudo-first-order and pseudo-second-order kinetics were applied. Furthermore, the Langmuir and Freundlich isotherm models were used to analyze the data. A comparison of adsorption capacities was made for different types of adsorbents. In comparison with the newly synthesized *M.Ti*_2_*CT_x_*, a certain amount of identically adsorbent including graphene oxide (GO), Ti_3_C_2_T_x_ MXene, or Fe_3_O_4_ was introduced into 15 mL of 10 ppm pollutant solution. The experiments were performed under optimized experimental conditions. Samples were taken at different time intervals, diluted 10-fold, and analyzed using ICP-MS.

To validate the usefulness of the synthesized materials for radioactive ions adsorption, 10 mg *M.Ti*_2_*CT_x_A_III_* adsorbent was introduced into different matrices, e.g., distilled water, tap water, and seawater. The seawater was simulated using 10,400 ppm Na^+^, 390 ppm K^+^, 1270 ppm Mg^2+^, and 405 ppm Ca^2+^ ions. The aliquot was drawn after 24 h reaction and analyzed using ICP-MS to determine the residual amount.

## 3. Results and Discussions

### 3.1. Characterization of *M.Ti_2_CT_x_-A_III_*

Ti_2_AlC MAX phase was successfully fabricated by high temperature (1350 °C) sintering of Al and TiC in a 1:2 ratio [36]. Further, the hydrothermal alkalization of Ti_2_AlC powder resulted in different types of magnetic structures. In a one-step hydrothermal treatment method, magnetic properties were induced during the alkalization process. To achieve the best magnetic properties and a well-exfoliated structure, synthesis conditions such as NaOH concentration (5–15 M), and treatment time (12–48 h) were varied; however, the temperature was kept unchanged at 200 °C). The obtained magnetic structures using the different synthesis conditions are shown in Table 1.

The synthesized magnetic structures were used for radioactive cations Sr^2+^ and Cs^+^ removal from water and the results revealed that *M.Ti*_2_*CT_x_-A_III_* showed the highest removal efficiencies for both Sr^2+^ and Cs^+^ ions. *M.Ti*_2_*CT_x_-A_III_* was synthesized by the hydrothermal treatment of Ti_2_AlC in the presence of 5 molar sodium hydroxide and FeSO_4_ at 200 °C for 48 h. Scanning electron microscopy showed different morphologies for the different exfoliated phases.

Figure 1a shows the SEM images of the layered Ti_2_AlC MAX phase. After hydrothermal treatment, Ti_2_AlC changed into 2D sheet-like structures as shown in Figure 1b,c. The growth of the nanostructure depends on the synthesis conditions; changes in the alkalinity and synthesis time led to different structures of the final product. For the synthesis of *M.Ti*_2_*CT_x_-A_III_*, 5 M NaOH and 250 mg of FeSO_4_·7H_2_O was sufficient to remove the layers of Al from Ti_2_AlC and induce maximum magnetization as corroborated by energy-dispersive X-ray spectroscopy (EDS) analysis and magnetic field-dependent magnetic measurements. Micron-sized Fe_3_O_4_ particles with octahedron geometry were also observed in SEM images confirming the formation of a composite material (Figure 1d). A higher concentration of NaOH (10–15 M) could not create any definitive structural shape and further caused a decrease in the observed magnetization. Further, EDS measurements established the complete elimination of the *Al* layers from Ti_2_AlC as shown in Appendix A; consequently, the newly developed structures were named *M.Ti*_2_*CT_x_*, where *T_x_* characterizes the surface functional groups such as –Na, –OH, and –O, and the presence of all elements in the EDS graph (Figure 1e,f) [35]. Additionally, the elemental mapping in SEM-EDS analysis (Figure 1e) of the samples after Sr^2+^ and Cs^+^ adsorption (Sr^2+^@*M.Ti*_2_*CT_x_-A_III_* or Cs^+^@*M.Ti*_2_*CT_x_-A_III_*) presented a uniform distribution of all characteristic elements, including the presence of significant amounts of Sr^2+^ and Cs^+^.

Based on the initial assessments of radioactive cation removal by materials and morphology examined by SEM data analysis, we selected only *M.Ti*_2_*CT_x_-A_III_* for further characteristic analyses. In comparison with other exfoliated structures, *M.Ti*_2_*CT_x_-A_III_* exhibited higher adsorption capacity for both Sr^2+^ and Cs^+^; thus, *M.Ti*_2_*CT_x_-A_III_* was selected for further physio-chemical characteristics studies. The XRD pattern of *M.Ti*_2_*CT_x_-A_III_* indicated the perseverance of crystalline structures after exfoliation at 200 °C and treatment with 5 M sodium hydroxide. After exfoliation, the intensity of the characteristic peak in Ti_2_AlC (2*θ* = 39.26°) decreased (red color spectrum Figure 2a). Furthermore, the peak at 2*θ* = 12.90° moved to 2*θ* = 9.98° in *M.Ti*_2_*CT_x_-A_III_* (Figure 2a). Fe_3_O_4_ was synthesized for reference, and the XRD pattern of Fe_3_O_4_ is shown in blue color in Figure 2a. Fe_3_O_4_ was synthesized thorough hydrothermal treatment using 5 M NaOH at 200 °C for 48 h. The representative peaks in the XRD pattern, 2*θ* = 35°, 30°, 39°, 57°, and 18.07° match with available literature and PDF reference code 01-089-0688. Further, the peak at 2*θ* = ~18° strongly suggests the octahedron morphology of Fe_3_O_4_ nanoparticles [37]. Therefore, the peaks at 2*θ* = ~18° and ~35° in *M.Ti*_2_*CT_x_-A_III_* (green color spectrum in Figure 2a) represent the emergence of magnetic Fe_3_O_4_ nanoparticles during the exfoliation process. The produced *M.Ti*_2_*CT_x_-A_III_* exhibited ferrimagnetic behavior, as evidenced by the magnetic field-dependent magnetization measurements. A characteristic magnetic hysteresis loop is shown in Figure 2b. At room temperature, *M.Ti*_2_*CT_x_-A_III_* showed a saturation magnetization of 10.69 emu/g, which is expectedly lower than that of pure Fe_3_O_4_ (52.02 emu/g). The reduction in magnetization is understandable as non-magnetic Ti_2_CT_x_ is present in large amounts in the magnetic composite *M.Ti*_2_*CT_x_-A_III_*.

The Brunauer–Emmett–Teller (BET) surface area of Ti_2_AlC MAX phase and *M.Ti*_2_*CT_x_-A_III_* were measured, and the results showed a sudden increase in the surface area from 0.618 to 29.33 m^2^/g for the parent MAX phase and *M.Ti*_2_*CT_x_-A_III_* composite, respectively (Appendix A). An unexpected and noteworthy rise in surface area in the after-exfoliation samples was potentially due to the elimination of Al layers and the formation of sheet-like structure in *M.Ti*_2_*CT_x_-A_III_*. Furthermore, in a Barrett–Joyner–Halenda (BJH) plot the mean pore diameter of *M.Ti*_2_*CT_x_-A_III_* was 32.04 nm (Appendix A). 

X-ray photoelectron spectroscopy (XPS) additionally revealed the formation of *M.Ti*_2_*CT_x_-A_III_* and changes in all the elements states (Appendix A). The complete spectra of *M.Ti*_2_*CT_x_-A_III_* showed the presence of representative elements including Ti 2p, O 1s, C 1s, Fe 2p, and Na 1s (bottom spectrum in Figure 3). Furthermore, after adsorption of the radionuclides, peaks corresponding to Sr 3d and Cs 3d emerged in the *M.Ti*_2_*CT_x_-A_III_* samples (blue color spectrum in Figure 3). The chemical changes that occurred in the different phases of adsorbent authenticate the successful syntheses of desired materials and loading of radionuclides onto materials. Furthermore, in regional peak fitting analysis of Sr 3d, two de-convoluted peaks were found at binding energies of 133.71 and 135.5 eV, which can be designated as Sr 3d_5/2_ and Sr 3d_3/2_ (Appendix A). Moreover, we have observed a radical decrease in Na 1s peak intensity after Sr^2+^ adsorption onto *M.Ti*_2_*CT_x_-A_III_*, which could be due to the ion exchange of Sr^2+^ with Na ions (Appendix A). Further, there was a decrease in peak intensity and peak shifting in C 1s after adsorption of Sr^2+^ and Cs^+^ onto *M.Ti*_2_*CT_x_-A_III_* (Appendix A). Further, in regional XPS data recording, we could not find the Cs 1s peak and this could be due to the presence of a comparatively small amount of Cs^+^ in the *M.Ti*_2_*CT_x_-A_III_*. However, a signal of Cs 3d was found at around 690 eV. Furthermore, SEM-EDS analysis of the Cs-laden *M.Ti*_2_*CT_x_-A_III_* sample exhibited the presence of a significant amount of Cs in it (1.52 Wt.%). Further details are given in Appendix A. 

### 3.2. Radionuclide Adsorption

#### 3.2.1. Adsorptive Behavior of *M.Ti*_2_*CT_x_*

The presence of radioactive nuclides such as Sr^2+^ and Cs^+^ in wastewater is a major risk and serious threat to humans and other living organisms. Therefore, before disposal, the removal of Cs^+^ and Sr^2+^ from nuclear waste is very crucial. This work aims to examine the adsorptive performance of the synthesized magnetic adsorbent for Cs^+^ and Sr^2+^. The synthesized magnetic *M.Ti*_2_*CT_x_* exhibited higher porosity, magnetic behavior, and the presence of surface functional groups such as ⎼Na, ⎼OH, ⎼O, and FeO. Structures with these properties could be used in the purification of water contaminated with heavy metal ions, especially cationic radionuclide removal from water. Therefore, the synthesized materials were tested against Sr^2+^ and Cs^+^ in batch adsorption tests, and the results are displayed in Table 2. The radionuclide adsorption from solution at certain concentrations (12.811 and 10.911 ppm for Cs^+^ and Sr^2+^, respectively) was determined for eight different materials in batch adsorption tests. The adsorbent named *M.Ti*_2_*CT_x_-A_III_* exhibited the highest efficiency for Cs^+^ and Sr^2+^. The maximum removal efficiency for Cs^+^ was 79.43%, which was higher than all other magnetite materials. Furthermore, in the case of Sr^2+^, all synthesized nanostructures exhibited excellent removal efficiency and *M.Ti*_2_*CT_x_-A_III_* exhibited more than 99% removal. The nanostructure showed a higher affinity for divalent cations Sr^2+^ with the highest removal efficiency between 96 and 99% as compared to monovalent radionuclides Cs^+^, which was between ~2 and 80%.

#### 3.2.2. Comparison with Other Materials

The *M.Ti*_2_*CT_x_-A_III_* nanostructure synthesized in this work was compared with other similar benchmark adsorbents, including 2D GO, 2D Ti_3_C_2_T_x_ MXene, Fe_3_O_4_, and *Alk-Ti*_2_*C_sheet_*. The synthesis method for Ti_3_C_2_T_x_ MXene is presented in our previous study [38]. The 2D GO nanosheets used in this work were produced by a modified Hummer’s method [39] and further details are also provided in our previous work [37]. The synthesis method for the reference Fe_3_O_4_ is also provided before in the material synthesis section. *Alk-Ti*_2_*C_sheet_* nanosheets were synthesized following the same procedure used for *M.Ti*_2_*CT_x_-A_III_* synthesis under different synthesis conditions. *Alk-Ti*_2_*C_sheet_* can be synthesized by treating 200 mg of Ti_2_AlC MAX phase in 5 M NaOH at 200 °C for 12 h [35]. In a comparison adsorption test, the *M.Ti*_2_*CT_x_-A_III_* showed the highest removal efficiency for Sr^2+^ (99.60%) as compared to Fe_3_O_4_, GO, and Ti_3_C_2_T_x_ MXene, which was 18.44, 94.70, and 16.60%, respectively (Figure 4a). For Cs^+^ removal, *M.Ti*_2_*CT_x_-A_III_* also performed well among all other completive materials except *Alk-Ti*_2_*C_sheet_*. The Cs^+^ removal efficiency was 73.40, 10.47, 33.68, and 37.49% for *M.Ti*_2_*CT_x_-A_III_*, Fe_3_O_4_, GO, and Ti_3_C_2_T_x_ MXene, respectively (Figure 4b). *Alk-Ti*_2_*C_sheet_* showed excellent adsorption efficiency for both Cs^+^ and Sr^2+^ (93.77%, and 99.68%, respectively) as compared to *M.Ti*_2_*CT_x_-A_III_*. The main reason for higher removal efficacy is the well-exfoliated and well-defined structure of *Alk-Ti*_2_*C_sheet_*. *Alk-Ti*_2_*C_sheet_* was synthesized in only NaOH-solution in the absence of iron sulphate and thus the exfoliation was more efficient, and a comparatively shorter time was required for the Al layer to be etched out from the Ti_2_AlC phase. The well-defined structure and functional groups played important roles in a greater number of nuclides cations becoming entrapped and being captured. However, without magnetic properties, it was very difficult to remove *Alk-Ti*_2_*C_sheet_* from the water after contact with radionuclides. On the other hand, *M.Ti*_2_*CT_x_-A_III_* offers easy separation by using an external magnet after the adsorption experiment. Therefore, *M.Ti*_2_*CT_x_-A_III_* could be a better alternative option for the easy separation of radionuclide-loaded nanomaterials, and thus the discharge of nanoparticles into the environment can be circumvented. The stated results showed that *M.Ti*_2_*CT_x_-A_III_* has comparatively higher binding abilities against Sr^2+^ and Cs^+^ as compared to other benchmark materials. A comparison between the fabricated *M.Ti*_2_*CT_x_-A_III_* and previously reported materials is illustrated in Table 3.

#### 3.2.3. Effect of Solution pH

The pH of a solution usually plays a crucial role in the adsorption of metal ion contaminants in liquid phase, as the removal is a pH-dependent process. Therefore, the influence of the solution’s pH on Sr^2+^ and Cs^+^ was assessed in this work. The adsorption efficiency of *M.Ti*_2_*CT_x_-A_III_* for Sr^2+^ and Cs^+^ was performed at various pH values ranging from pH = 2 to 9. The findings revealed that at pH = 2, both Sr^2+^ and Cs^+^ adsorption were as low as 60.58 and 16.40%, respectively, but increased significantly at pH values ranging from 3 to 9 to 99.97 and 60.0%, respectively (Figure 4c). The increase in removal affinity evidently demonstrated the influence of Sr^2+^ and Cs^+^ ionic form and also the surface characteristics of the *M.Ti*_2_*CT_x_-A_III_* material used. *M.Ti*_2_*CT_x_-A_III_* demonstrated low removal efficiency at lower pH of the solution, where the surface charges on *M.Ti*_2_*CT_x_-A_III_* were reduced, perhaps due to the competition of positive charges and M^+^ ions [25]. Hydroxyl group protonation on *M.Ti*_2_*CT_x_-A_III_* is possibly a cause for this, as it produces repulsive forces in highly acidic media with very low pH. These results strongly indicate that the adsorption of Sr^2+^ and Cs^+^ ions onto *M.Ti*_2_*CT_x_-A_III_* is a pH-dependent phenomenon.

#### 3.2.4. Effect of Contact Time

The synthesized *M.Ti*_2_*CT_x_-A_III_* nanostructures demonstrated exceptional adsorption behavior with fast sorption kinetics for Sr^2+^ and Cs^+^ ions. For the adsorption kinetics test, a binary solution containing Sr^2+^ and Cs^+^ ions was prepared, and a certain amount of *M.Ti*_2_*CT_x_-A_III_* was introduced into the solution and agitated for 12 h. The results showed that >97% of total Sr^2+^ (8.55 ppm) was adsorbed in just 15 min and adsorption equilibrium was achieved in 1 h. In the case of Cs^+^, the adsorption process was also fast, as, in the first 15 min, ~68% Cs^+^ (7.91 ppm) was adsorbed and achieved equilibrium state in 1 h. Further, sorption kinetics models such as the Lagergren-pseudo-first order and second-order adsorption kinetics models were applied on obtained sets of data to obtain insight about fast kinetics and possible interaction mechanism between solid–liquid phases of the adsorbent and adsorbate (Figure 5a,b). Among the applied kinetic models of adsorption, the pseudo-second-order kinetic model fitted very well with the data sets of Sr^2+^@*M.Ti*_2_*CT_x_-A_III_* and Cs^+^@*M.Ti*_2_*CT_x_-A_III_* as compared to the pseudo-first-order model. This rapid adsorption of radionuclide ions may be due to the large surface area and porosity and highly occupied empty binding sites on the *M.Ti*_2_*CT_x_-A_III_* [50]. Moreover, the calculated equation parameters were close to the experimental results and validated the process of adsorption. The adsorption capacity (*Q_t_*) calculated by second-order kinetics for Sr^2+^ and Cs^+^ was 12.78 and 9.15 mg/g, respectively, which was close to the experimental adsorption density of 12.77 and 8.99 mg/g, respectively. Additionally, the regression coefficient value (*R*^2^) was 1.0 and 0.999 for Sr^2+^ and Cs^+^, respectively. Thus, the above results obtained from the kinetic test indicated that the adsorption of both Sr^2+^ and Cs^+^ onto *M.Ti*_2_*CT_x_-A_III_* was a chemical interaction with a rate-limiting step.

#### 3.2.5. Adsorption Isotherm

Adsorption isotherm models were further assessed to obtain an understanding of the adsorption of radionuclides. Separate sets of adsorption experiments were performed for each nuclide’s cations at different initial concentrations. The maximum adsorption densities of *M.Ti*_2_*CT_x_**-A_III_* for Sr^2+^ and Cs^+^ at saturation point were 357.60 and 140.42 mg/g, respectively. Adsorption isotherm models, such as Langmuir and Freundlich isotherms, were applied to the experimentally obtained data (Table 4). The Langmuir isotherm model with higher regression coefficient (*R*^2^) values fitted well as compared to the Freundlich isotherm model (Figure 5c,d). The Langmuir isotherm model, with calculated maximum adsorption capacities of 376.05 and 142.88 mg/g for Sr^2+^ and Cs^+^, respectively, indicate that the radionuclides were adsorbed as a monolayer on *M.Ti*_2_*CT_x_**-A_III_*. The adsorption capacities calculated from the Langmuir isotherm were close to the obtained experimental values (Table 4). 

#### 3.2.6. Practical Application of Alk-Ti_2_C_sheet_

In nuclear power plants, management of nuclear waste is imperative. Therefore, the capability of the synthesized magnetic nanostructures for nuclear waste treatment was evaluated to establish their efficacy. Accordingly, the adsorption of Cs^+^ and Sr^2+^ cations in DI, tap, and seawater, filled with coexisting ions, was analyzed. The competing ions, such as Ca^2+^, Mg^2+^, K^+^, and Na^+^, were inserted, with concentrations similar to the matrices. The removal efficiencies in tap water and simulated seawater were very good compared to the control experiments. The results obtained from the experiments are illustrated in Table 5. The adsorption of Sr^2+^ and Cs^+^ was performed in both single and binary (Sr^2+^ + Cs^+^) solution, Sr^2+^ exhibiting excellent removal efficiency in all matrices. The Cs^+^ removal efficiency was lower than that of Sr^2+^; furthermore, in DI water, the adsorption efficiency was ~93% but reduced to ~70 and ~31% in tap and seawater, respectively. The removal efficiencies were lower in seawater, which is due to the presence of competitive cations in simulated seawater. The higher concentration of Mg^2+^ and Ca^2+^ could be responsible for the decrease in Sr^2+^, and Na^+^ and K^+^ might influence the Cs^+^ adsorption. The results showed a higher affinity for divalent cations over single-valent cations, as we experienced in previous experimental tests. Overall, the results established remarkable adsorption efficiency of *M.Ti*_2_*CT_x_**-A_III_*.

## 4. Conclusions

In this work, we have performed etching of the Ti_2_AlC phase using a green hydrothermal alkalization process to etch out the Al layer. The magnetic properties were successfully incorporated during A-layer etching. The resulting *M.Ti*_2_*CT_x_**-A_III_* exhibited sheet-like morphology with abundant surface-terminal groups. The said characteristics and unique morphology marked it as an outstanding material for Sr^2+^ and Cs^+^ adsorption. *M.Ti*_2_*CT_x_**-A_III_* was able to adsorb Sr^2+^ and Cs^+^ swiftly and efficiently in numerous matrices with very high removal efficiencies, including deionized, tap, and seawater. In the selective removal test, the fast and excellent adsorption capacity of Cs^+^ and Sr^2+^ in seawater (325.59 and 1014.02 µg/g, respectively) validates the potential of the synthesized material for practical applications. The radionuclide Sr^2+^ and Cs^+^ removal procedure was dependent on the pH of the solution, monolayer adsorption process, and rate-limiting parameters, and the maximum adsorption capacities of *M.Ti*_2_*CT_x_**-A_III_* was 376.05 and 142.88 mg/g for Sr^2+^ and Cs^+^, respectively. These findings suggest that the etching of Ti_2_AlC by adopting fluoride-free procedure might be an unconventional but feasible approach to preparing nanomaterials for environmental applications. This research also advances the use of innovative 2D nanomaterials to address radioactive waste remediation challenges.

## Figures and Tables

**Figure 1 nanomaterials-12-03253-f001:**
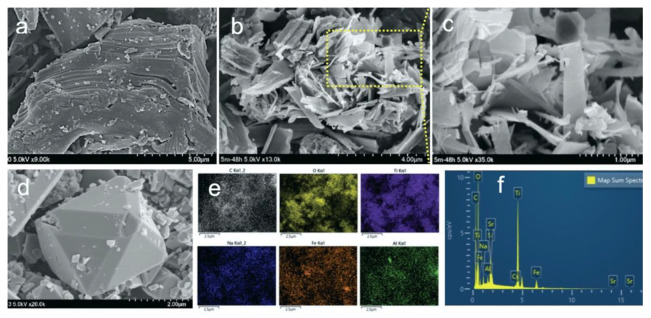
SEM image of (**a**) Ti_2_AlC MAX phase, and (**b**,**c**) magnetic *M.Ti*_2_*CT_x_-A_III_* nanostructures at different magnification, (**d**) Fe_3_O_4_ magnetite, (**e**) elemental mapping of magnetic *M.Ti*_2_*CT_x_-A_III_* after Sr^2+^ and Cs^+^ adsorption (Sr^2+^, Cs^+^@ *M.Ti*_2_*CT_x_-A_III_*), and (**f**) EDS elemental mapping of Sr^2+^, Cs^+^@ *M.Ti*_2_*CT_x_-A_III_*.

**Figure 2 nanomaterials-12-03253-f002:**
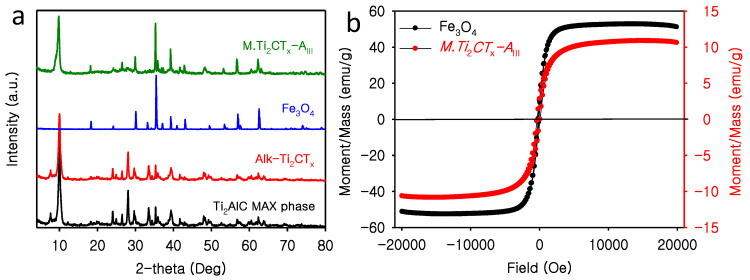
(**a**) XRD diffraction pattern of different phases of the fabricated materials and (**b**) magnetic field-dependent magnetization curves of *M.Ti*_2_*CT_x_-A_III_* along with that of the Fe_3_O_4_ reference.

**Figure 3 nanomaterials-12-03253-f003:**
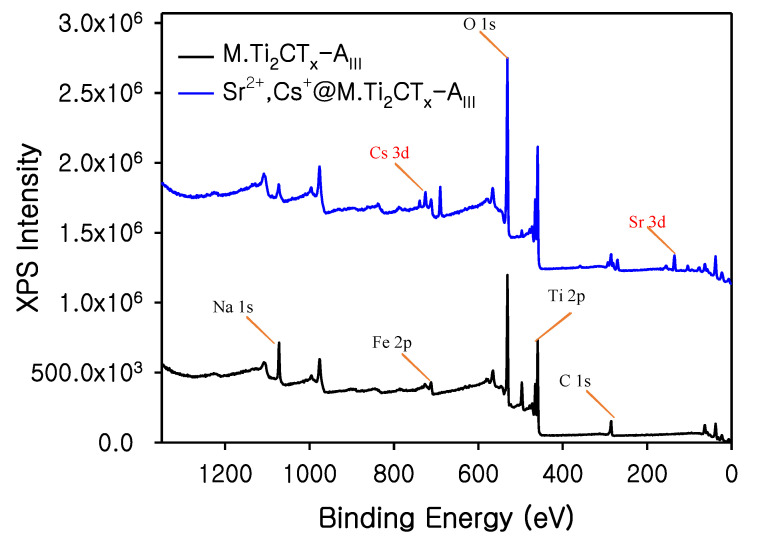
X-ray photoelectron spectroscopy of *M.Ti*_2_*CT_x_-A_III_* before and after adsorption of radionuclides.

**Figure 4 nanomaterials-12-03253-f004:**
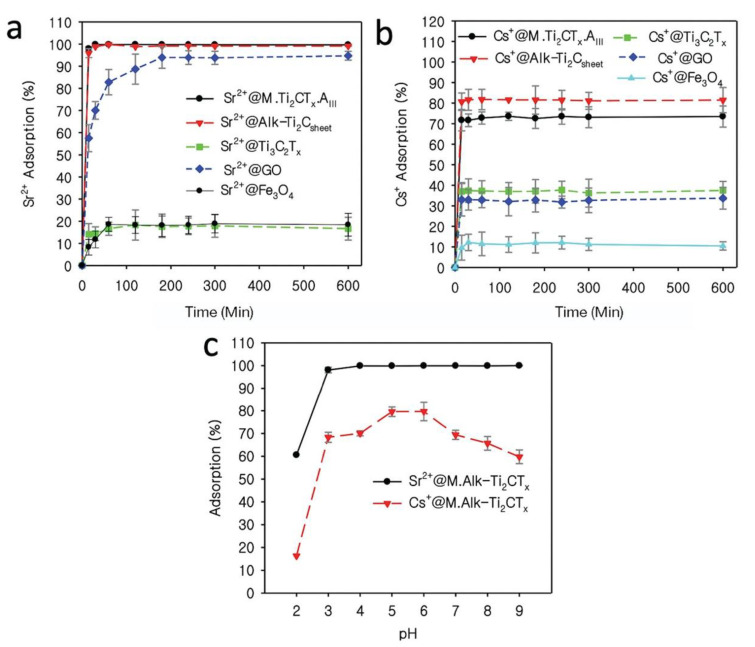
Comparison of adsorption efficiency of different materials for (**a**) Sr^2+^ and (**b**) Cs^+^ in deionized water. *M.Ti*_2_*CT_x_-A_III_* exhibited exceptionally high removal capacity for both Sr^2+^ and Cs^+^ radionuclides. (**c**) Effect of solution pH on Sr^2+^ and Cs^+^ adsorption by *M.Ti*_2_*CT_x_-A_III_*.

**Figure 5 nanomaterials-12-03253-f005:**
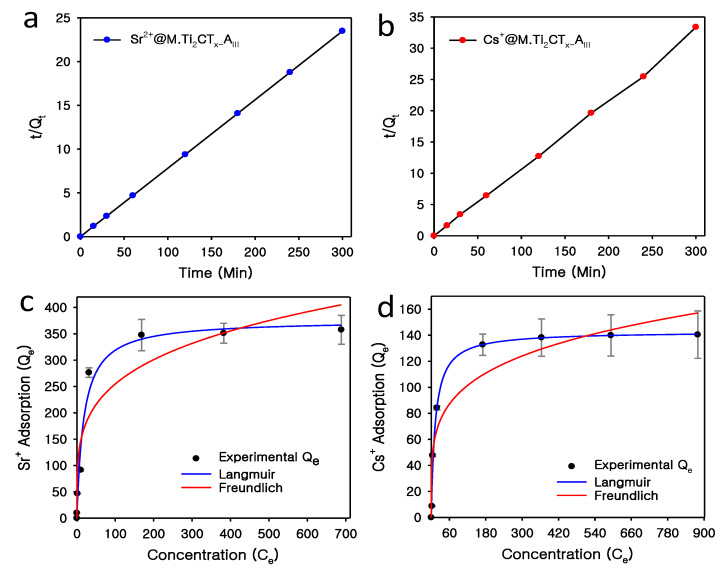
Adsorption kinetics: pseudo-second-order kinetics graphs of (**a**) Sr^2+^@*M.Ti*_2_*CT_x_-A_III_* and (**b**) Cs^+^@*M.Ti*_2_*CT_x_-A_III_*. Langmuir and Freundlich adsorption isotherm model graphs of (**c**) Sr^2+^ and (**d**) Cs^+^ adsorbed by *M.Ti*_2_*CT_x_-A_III_*.

**Table 1 nanomaterials-12-03253-t001:** *M.Ti*_2_*CT_x_* hybrid nanostructures synthesized using Ti_2_AlC and FeSO_4_.7H_2_O at different NaOH concentrations, temperatures, and times.

No.	Material Code	Synthesis Conditions
1	*M.Ti* _2_ *CT_x_-A_I_*	Synthesized at 5 M NaOH, 200 °C for 12 h
2	*M.Ti* _2_ *CT_x_-A_II_*	Synthesized at 5 M NaOH, 200 °C for 24 h
3	*M.Ti* _2_ *CT_x_-A_III_*	Synthesized at 5 M NaOH, 200 °C for 48 h
4	*M.Ti* _2_ *CT_x_-B_I_*	Synthesized at 10 M NaOH, 200 °C for 12 h
5	*M.Ti* _2_ *CT_x_-B_II_*	Synthesized at 10 M NaOH, 200 °C for 24 h
6	*M.Ti* _2_ *CT_x_-B_III_*	Synthesized at 10 M NaOH, 200 °C for 48 h
7	*M.Ti* _2_ *CT_x_-C_I_*	Synthesized at 15 M NaOH, 200 °C for 12 h
8	*M.Ti* _2_ *CT_x_-C_II_*	Synthesized at 15 M NaOH, 200 °C for 24 h
9	*M.Ti* _2_ *CT_x_-C_III_*	Synthesized at 15 M NaOH, 200 °C for 48 h

**Table 2 nanomaterials-12-03253-t002:** Sr^2+^ and Cs^+^ removal by various *M.Ti*_2_*CT_x_* nanostructures.

Type of Adsorbent	Cs^+^ Concentration (ppm)		Removal (%)	Sr^2+^ Concentration (ppm)		Removal (%)
	Initial	Final		Initial	Final	
*M.Ti* _2_ *CT_x_-A_I_*	12.181	3.290	72.99	10.911	0.006	99.94
*M.Ti* _2_ *CT_x_-A_II_*	12.181	3.022	75.19	10.911	0.011	99.89
*M.Ti* _2_ *CT_x_-A_III_*	12.181	2.505	79.43	10.911	0.009	99.91
*M.Ti* _2_ *CT_x_-B_I_*	12.181	4.857	60.12	10.911	0.007	99.93
*M.Ti* _2_ *CT_x_-B_II_*	12.181	8.643	29.05	10.911	0.022	99.79
*M.Ti* _2_ *CT_x_-B_III_*	12.181	9.484	22.14	10.911	0.022	99.80
*M.Ti* _2_ *CT_x_-C_I_*	12.181	9.752	19.94	10.911	0.048	99.56
*M.Ti* _2_ *CT_x_-C_III_*	12.181	12.017	1.35	10.911	0.434	96.02

**Table 3 nanomaterials-12-03253-t003:** Comparison between *M.Ti*_2_*CT_x_-A_III_* and other benchmark materials for maximum adsorption capacity of Sr^2+^ and Cs^+^ adsorption.

Adsorbent	Radionuclide	Adsorption Capacity (mg/g)	Effective pH	References
magnetite−silica composite	Sr^2+^ Cs^+^	343.48 93.42	7 6	[40]
OMt/alginate	Sr^2+^	42.41	6–11	[41]
PB-MHBs-3	Cs^+^	41.15	7	[10]
MgAl-LDH/GO	Sr^2+^	213.35	4–10	[42]
PB/Fe_3_O_4_/GO	Cs^+^	55.56	7	[43]
PB + CNTs	Cs^+^	142.85	4–8	[44]
RAFT-IIP	Sr^2+^	145.77	6–8	[45]
BaSO4/rGO	Sr^2+^	129.37	7–11	[46]
RGO/WO3	Sr^2+^	149.56	4–11	[47]
GO	Sr^2+^	140	6	[48]
GO-EDTA	Sr^2+^	158	6	
Magnetic Nb-CST	Sr^2+^ Cs^+^	14.38 11.18	9 4	[49]
*M.Ti* _2_ *CT_x_-A_III_*	Sr^2+^ Cs^+^	376.05 142.88	3–11 6	This work

**Table 4 nanomaterials-12-03253-t004:** Langmuir and Freundlich adsorption isotherm parameters for Sr^2+^ and Cs^+^ adsorption on *M.Ti*_2_*CT_x_-A_III_*.

Isotherm Model	Parameters	Values	
		Sr^2+^@*M.Ti*_2_*CT_x_-A_III_*	Cs^+^@*M.Ti*_2_*CT_x_-A_III_*
Langmuir (*Q_e_ = Q_m_K_a_C_e_*/1 + *K_a_C_e_*)	q_max_ (mg/g)	376.05	142.88
	k_L_	0.056	0.077
	*R* ^2^	0.976	0.987
Freundlich (*Q_e_* = *K_F_C_e_^(1/n)^*)	k_F_ (mg/g)	84.29	35.46
	1/n	0.240	1.029
	*R* ^2^	0.882	0.892

**Table 5 nanomaterials-12-03253-t005:** Radionuclide Sr^2+^ and Cs^+^ removal in various matrices using the synthesized magnetic adsorbent.

Radionuclide	Parameters	Matrices					
		Single-Element Solution			(Binary Solution: Sr^2+^ + Cs^+^)		
		Deionized Water	Tap Water	Simulated Seawater	Deionized Water	Tap Water	Simulated Seawater
Sr^2+^@ *M.Ti*_2_*CT_x_**-A_III_*	Initial Conc. (µg/L)	1100.23	1100.23	1100.23	1005	1005	1005
	Final Conc. (µg/L)	12.37	1.59	86.21	15.09	1.53	105.08
	Removal (Conc. (µg/g)	1087.87	1098.64	1014.02	989.91	1003.47	899.92
	Removal (%)	98.88	99.86	92.16	98.50	99.85	89.54
Cs^+^@ *M.Ti*_2_*CT_x_**-A_III_*	Initial Conc. (µg/L)	1042.67	1042.67	1042.67	998.87	998.87	998.87
	Final Conc. (µg/L)	58.04	310.85	717.071	61.41	323.91	886.75
	Removal (Conc. (µg/g)	984.63	731.81	325.59	938.52	674.96	112.12
	Removal (%)	94.43	70.19	31.23	93.85	67.57	11.22

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
