# Peer review of "A Hydrofluoric Acid-Free Green Synthesis of Magnetic M.Ti2CTx Nanostructures for the Sequestration of Cesium and Strontium Radionuclide"

_nanomaterials, 2022, doi:10.3390/nano12183253_

Round 1
Reviewer 1 Report
The work of J. Iqbal et al. focuses on environmentally friendly preparation of novel Fe-doped Ti2C MXene type materials developed for removal of radioactive Sr and Cs cations from radionuclide contaminated wastewaters. Both the concept (introduction of a novel layered material as adsorbent) and the purpose (elimination of dangerous substances from wastewater) are important and interesting, thus the work may be relevant for the readers of Nanomaterials. However, the manuscript in its present form contains serious shortcomings, thus publication without significant reworking cannot be recommended.
A) First of all, the characterization of the adsorbents created by alkali etching of the parent Ti2AlC MAX phase is incomplete. When assessing the performance of an adsorbent, its specific surface area (SSA) and surface state (i.e. surface composition, chemical state/bonding environment of the surface species, functional groups, etc.) are key parameters. Unfortunately, no data on any of them is shown. In particular, if the work tries to optimize the behavior of the created materials by varying the concentration of the etchant and the etching time, seeking connections between the synthesis parameters and the resulting structures cannot be neglected. Therefore, I recommend to determine the SSA and pore size distribution of some of the prepared compounds by nitrogen adsorption and discuss the results in terms of the synthesis parameters and functionality. Another important but missing feature of the produced MXene-like materials is their composition. Table 1 should be accompanied by data on bulk composition. For example, from Figure 1 it appears that Al remains and Na becomes quite homogeneously dispersed in the synthesis product, therefore, the level of Al removal and Na incorporation may be particularly interesting. Is there any relationship between these features, the synthesis parameters and the functionality of the materials? From the point of view of the planned application, the surface composition and the chemical state of the surface species are particularly important. This information should be extracted from the XPS measurements.
B) The nomenclature and identification of the synthesized materials is sometimes quite confusing. At first it appears that all studied materials are listed in Table 1. Then in Fig. 2a a diffractogram of a certain Alk-Ti2CTx appears, which is not explained either in the caption or the text. In sect 3.2.2 Alk-Ti2Csheet is mentioned as comparison partner, while in Figure 4 M.Alk-Ti2CTx is shown. Table 1 denotes the synthesized materials as M.Ti2C-A/B/C but at other places these materials are named as M.Ti2CTx-A/B/C. I believe in Table 1 the denomination and synthesis of all studied materials should be given and the denomination should be used in consequent manner.
C) The manuscript contains numerous smaller but embarrassing errors which also should be eliminated during revision. (i) the English usage, especially the sentence structures should be very carefully checked and corrected. (ii) experimental details of the XPS experiment should be given. (iii) line 209: what is "supermagnetic" behaviour? (iv): sect. 3.2.4: please explain to what equation data in Fig. 5a and b are fitted and give the appropriate reference. Please explain the relevance of the result. Similarly, in sect. 3.2.5 please explain why it is important that the adsorption isotherms can be fitted by the Langmuir model.
Taking into account the above impressions, I believe this work should be reconsidered for publication in Nanomaterials but a major revision is definitely needed before acceptance.
Author Response
Referee: 1
General Comment: The work of J. Iqbal et al. focuses on environmentally friendly preparation of novel Fe-doped Ti2C MXene type materials developed for removal of radioactive Sr and Cs cations from radionuclide contaminated wastewaters. Both the concept (introduction of a novel layered material as adsorbent) and the purpose (elimination of dangerous substances from wastewater) are important and interesting, thus the work may be relevant for the readers of Nanomaterials. However, the manuscript in its present form contains serious shortcomings, thus publication without significant reworking cannot be recommended.
Response: Authors are thankful to the reviewer for acknowledging the concept, purpose, and importance of this research work. We are also thankful for critically reviewing the manuscript. Most comments seemed to be fair and reasonable, so we paid heed to the reviewer’s advice and suggestions, and the manuscript has been massively revised. We are confident that the new version of the manuscript is well improved, and the reviewer will be satisfied with it.
Comment A: First of all, the characterization of the adsorbents created by alkali etching of the parent Ti2AlC MAX phase is incomplete. When assessing the performance of an adsorbent, its specific surface area (SSA) and surface state (i.e. surface composition, chemical state/bonding environment of the surface species, functional groups, etc.) are key parameters. Unfortunately, no data on any of them is shown. In particular, if the work tries to optimize the behavior of the created materials by varying the concentration of the etchant and the etching time, seeking connections between the synthesis parameters and the resulting structures cannot be neglected. Therefore, I recommend to determine the SSA and pore size distribution of some of the prepared compounds by nitrogen adsorption and discuss the results in terms of the synthesis parameters and functionality. Another important but missing feature of the produced MXene-like materials is their composition. Table 1 should be accompanied by data on bulk composition. For example, from Figure 1 it appears that Al remains and Na becomes quite homogeneously dispersed in the synthesis product, therefore, the level of Al removal and Na incorporation may be particularly interesting. Is there any relationship between these features, the synthesis parameters and the functionality of the materials? From the point of view of the planned application, the surface composition and the chemical state of the surface species are particularly important. This information should be extracted from the XPS measurements.
Response: The authors are thankful to the reviewer for these valuable recommendations. As recommended by the reviewer, The BET surface measurement and pore size distribution were arrived at and the results are incorporated into the relevant section of the manuscript. Figure S3a and S3b showed the BET adsorption/desorption isotherm and pore size distribution plots.
Furthermore, we completely agreed with the reviewer regarding the in-depth investigation of XPS analysis. Therefore, now we have included the results of regional peak fitting analysis for elements particularly adsorbed strontium and cesium. The results are shown in Figure S4 and the removal mechanism of radionuclides has been extended and discussed in detail from the XPS perspective. Furthermore, the bulk composition of M.Ti2CTx.AIII composite is given in Table S1 and deserved in detail in the relevant section of the manuscript.
Comment B: The nomenclature and identification of the synthesized materials is sometimes quite confusing. At first it appears that all studied materials are listed in Table 1. Then in Fig. 2a a diffractogram of a certain Alk-Ti2CTx appears, which is not explained either in the caption or the text. In sect 3.2.2 Alk-Ti2Csheet is mentioned as comparison partner, while in Figure 4 M.Alk-Ti2CTx is shown. Table 1 denotes the synthesized materials as M.Ti2C-A/B/C but at other places these materials are named as M.Ti2CTx-A/B/C. I believe in Table 1 the denomination and synthesis of all studied materials should be given and the denomination should be used in consequent manner.
Response: We are thankful to the reviewer for highlighting these nomenclature mistakes. Now we have corrected the nomenclature errors in Table 1 and all other sections of the manuscripts. Alk-Ti2CTx was synthesized as reference materials and the synthesis detail has now been added to the “2. Materials and Methods” section. Furthermore, the adsorption capacity of the Alk-Ti2Csheet for both Sr2+and Cs+ has now been added in Figures 4a and 4b.
Comment C: The manuscript contains numerous smaller but embarrassing errors which also should be eliminated during revision. (i) the English usage, especially the sentence structures should be very carefully checked and corrected. (ii) experimental details of the XPS experiment should be given. (iii) line 209: what is "supermagnetic" behaviour? (iv): sect. 3.2.4: please explain to what equation data in Fig. 5a and b are fitted and give the appropriate reference. Please explain the relevance of the result. Similarly, in sect. 3.2.5 please explain why it is important that the adsorption isotherms can be fitted by the Langmuir model.
Taking into account the above impressions, I believe this work should be reconsidered for publication in Nanomaterials but a major revision is definitely needed before acceptance.
Response: As recommended by the reviewer, we have addressed the concerns including (i) the improvement in English writing, grammar, and sentence structure, (ii) experimental detail of XPS has now been provided, and (iii) the word “supermagnetic” has been replaced with “superparamagnetic” behavior. (iv) An explanation has been incorporated regarding equation data in Figures 5a and 5b and the Langmuir isotherm models. The manuscript has now been revised massively and we are confident that the revision version is more comprehensive, and clearer, and will satisfy the expectations of the reviewers and editor.
Reviewer 2 Report
1. Would the results of testing water solutions of strontium and cesium be comparable to their dissolution in seawater? What is the difference in the effect of immobilization?
2. Why is the removal efficiency greater for strontium than with cesium? Is there some transport mechanism or is it just an electrostatic reason?
3. line 218 and Fig.3: I suggest that in the future, when you talk about the presence of Sr 3d and Cs 3d spectral lines, you also do a detailed XPS spectrum for Sr 3d and Cs 3d. The survey spectrum (Fig 3.) does not tell about the chemical bond and valence state of strontium and cesium.
Author Response
Referee: 2
Comment 1: Would the results of testing water solutions of strontium and cesium be comparable to their dissolution in seawater? What is the difference in the effect of immobilization?
Response: We are thankful to the reviewer for raising this important question. It is very important to evaluate the adsorbent efficacy of radioactive isotopes in different matrices. In seawater, both Sr2+ and Cs+ removal is quite challenging due to the higher competition between radionuclides and higher concentrations of salts. In this study, the removal of Sr2+ remains very high in the presence of higher concentrations of competing cations including Mg and Ca. However, due to the presence of monovalent Na+ ion, Cs+ removal and cause are low as compared to divalent Sr2+. So, when there is a huge no. competing cations' presence such low adsorption can be expected for Cs. Thus, Sr2+ ions have a tendency to immobilized in higher numbers as compared to Cs+, where Na+ will inhabit the immobilization of monovalent cations.
Comment 2: Why is the removal efficiency greater for strontium than with cesium? Is there some transport mechanism or is it just an electrostatic reason?
Response: For both Sr2+ and Cs+ adsorption, the functional groups such as Na, O, and OH present in M.Ti2CTx-AIII composites have played an integral role. For instance, Sr2+ exchange with Na+ and Cs+ react with O- through electrostatic interaction to form Cs2O. Furthermore, due to the presence of Na+ is monovalent ion comes in competition with Cs+ and causes low removal efficiency as compared to divalent Sr2+.
Comment 3: line 218 and Fig.3: I suggest that in the future, when you talk about the presence of Sr 3d and Cs 3d spectral lines, you also do a detailed XPS spectrum for Sr 3d and Cs 3d. The survey spectrum (Fig 3.) does not tell about the chemical bond and valence state of strontium and cesium.
Response: As recommended by the reviewer, the XPS regional peak fitting analysis was conducted and the results are incorporated in the “3.1 Characterization of M.Ti2CTx-AIII” section of the manuscript and the Figure S2 in supporting information.
Round 2
Reviewer 1 Report
The manuscript was significantly improved in this revision round. However, unfortunately, my issue with the incomplete structural characterization was only partially addressed. Therefore, I cannot recommend the work for acceptance until certain details are further elaborated.
As the starting material for synthesis was Ti2AlC (Ti:C ratio: 1:0.5), the large abundance of C in the product (Ti:C ratio around 1:3-1:5 from Tables S1 and S2) requires a short discussion. Does it mean that a significant part of Ti was dissolved during the etching?
In my previous review I recommended specific surface area determinations. Maybe I did not stress enough but I would have wanted to see a comparison of the SSA e.g. for materials prepared by the A, B and C or the I, II, and III routes to check whether they are similar or some SSA differences can explain the differences in the absorption behavior. The increase of the SSA after exfoliation is by no means surprising and I assume that further optimization of the materials (if planned in the future) could be attempted at least partly in the direction of increasing their SSA.
Exactly the same could be told about the XPS analysis. I hoped for a comparison of different materials (e.g. those with good and bad Cs removal ability). I still cannot find the details of the XPS measurement in Sect. 2.2. The extent of analysis of the XPS results given in the manuscript is not satisfactory. A table with surface concentration derived from XPS is missing. An attempt to identify the chemical state of the components (e.g. C: carbidic? graphitic? hydrocarbon-like?, Ti: oxidized? carbidic?, Na, Sr, Cs: ionic? etc.) should also be shown. Two minor comments: the Sr 3d spectrum has little added value in itself and as far as Cs is concerned, its 1s peak cannot be measured with laboratory X-ray sources while the Cs 3d doublet is expected somewhere around 726 and 740 eV binding energies and checking for Cs 4d around 80 eV may also be useful. At the same time, the note about probable Sr-Na exchange is instructive, which indicates the importance of XPS in such investigations.
Nevertheless, considering (i) the intended message and completeness of the manuscript, (ii) the fact that the XPS analysis in its present form is not acceptable and (iii) the fact that an in-depth XPS study of several more samples could be indeed time-consuming at this point, I recommend the Authors to sacrifice some of the scientific merit of the work and either remove completely the XPS part or reduce it only to the mentioning of the observation of Sr and Cs on the surface after the adsorption experiment. In the latter case a table with composition data (and maybe the present Figure 3) in the Supplementary material is needed.
In addition, there remained a few small issues in the manuscript which can be resolved with little effort:
Lines 234-238: the two sentences are redundant.
Line 452: fluoride-free?
Supporting information: what is the source of Si in Table S2? To what environment was this sample exposed?
Author Response
Response to the reviewers’ comments and list of modifications (Round-II)
Manuscript Reference: Nanomaterials-1871641
Title: A hydrofluoric acid-free green synthesis of magnetic M.Ti2CTx nanostructures for the sequestration of cesium and strontium radionuclide
Dear Editor,
We are thankful to the editor and reviewers for giving their valuable time to our manuscript. We would also like to thank the reviewers for their valuable comments and in-depth review. In this round of review, the reviewers have highlighted some important issues, and their inputs are very helpful for further improving this research article. We hope that the editor and reviewers will find our responses and changes in manuscript satisfactory. Please, find below the reviewers’ comments and our responses inserted after each comment (in red).
Referee: 1
The manuscript was significantly improved in this revision round. However, unfortunately, my issue with the incomplete structural characterization was only partially addressed. Therefore, I cannot recommend the work for acceptance until certain details are further elaborated.
As the starting material for synthesis was Ti2AlC (Ti:C ratio: 1:0.5), the large abundance of C in the product (Ti:C ratio around 1:3-1:5 from Tables S1 and S2) requires a short discussion. Does it mean that a significant part of Ti was dissolved during the etching?
Response: We authors are thankful to the reviewer for his/her critical review of our manuscript. We have revised the manuscript as suggested by the reviewer and we are hopeful that the modifications made in the new version will meet the reviewer's expectations.
Regarding Ti: C ratios in the M.Ti2CTx-AIII sample, the SEM-EDS analysis only gives surface measurements of elements which is not trustworthy. However, are also agreed with the reviewer that due to the very high alkalinity and high temperature, the MAX phase structure deformed and some Ti may release. For comparison, the elemental composition measured in XPS analysis is more accurate which is 21.89: 16 atom% for Ti: C.
Comment: In my previous review I recommended specific surface area determinations. Maybe I did not stress enough but I would have wanted to see a comparison of the SSA e.g. for materials prepared by the A, B and C or the I, II, and III routes to check whether they are similar or some SSA differences can explain the differences in the absorption behavior. The increase of the SSA after exfoliation is by no means surprising and I assume that further optimization of the materials (if planned in the future) could be attempted at least partly in the direction of increasing their SSA.
Response: We are thankful to the reviewer for these essential suggestions, however, due to unavailability, experimental step-up, and time limitation we are unable to perform SSA measurements for all the samples. However, we will consider it in our future work.
Comment: Exactly the same could be told about the XPS analysis. I hoped for a comparison of different materials (e.g. those with good and bad Cs removal ability). I still cannot find the details of the XPS measurement in Sect. 2.2. The extent of analysis of the XPS results given in the manuscript is not satisfactory. A table with surface concentration derived from XPS is missing. An attempt to identify the chemical state of the components (e.g. C: carbidic? graphitic? hydrocarbon-like?, Ti: oxidized? carbidic?, Na, Sr, Cs: ionic? etc.) should also be shown. Two minor comments: the Sr 3d spectrum has little added value in itself and as far as Cs is concerned, its 1s peak cannot be measured with laboratory X-ray sources while the Cs 3d doublet is expected somewhere around 726 and 740 eV binding energies and checking for Cs 4d around 80 eV may also be useful. At the same time, the note about probable Sr-Na exchange is instructive, which indicates the importance of XPS in such investigations.
Nevertheless, considering (i) the intended message and completeness of the manuscript, (ii) the fact that the XPS analysis in its present form is not acceptable and (iii) the fact that an in-depth XPS study of several more samples could be indeed time-consuming at this point, I recommend the Authors to sacrifice some of the scientific merit of the work and either remove completely the XPS part or reduce it only to the mentioning of the observation of Sr and Cs on the surface after the adsorption experiment. In the latter case a table with composition data (and maybe the present Figure 3) in the Supplementary material is needed.
Response: The authors are thankful to the reviewers, for this important comment. As suggested by the reviewer, the table of surface composition is given in Table S3. We performed XPS measurements twice and still, we were unable to detect the presence of Cs and Fe in surface composition which could be due to the overlapping of other elements or the presence of a very small portion of both Fe and Cs in composition to other elements which are present in bulk. However, in the full scan XPS spectrum, peaks of Fe 2p and Cs 3d were detected. In addition, the SEM-EDS analysis confirmed the presence of both Fe Cs and VSM analysis magnetic measurements confirmed the ferromagnetic behavior which is obviously due to the presence of Fe3O4 particles in the M.Ti2CTx composite.
The X-ray scanning at 80 eV for Cs 4d is an imported suggestion and we will consider it in our next work. We have investigated the Sr-Na, and Sr-O interaction in-depth and the results are incorporated in the relevant section of the manuscript, and peak fitting graphs of C 1s, Na 1s, and O 1s are illustrated in Figure S2. We also agreed with the reviewer's point of view regarding XPS analysis of many samples which is practically not possible at this time. Therefore, we would like to keep shorten the XPS analysis and present an overview of XPS results in this work.
In addition, there remained a few small issues in the manuscript which can be resolved with little effort:
Lines 234-238: the two sentences are redundant.
Response: The mentioned error has been corrected.
Line 452: fluoride-free?
Response: MAX phases are usually etched by using hydrofluoric acid which could cause the release of fluoride into the environment. Therefore, a fluoride-free technique such as hydrothermal alkaline treatment could be an alternative process.
Supporting information: what is the source of Si in Table S2? To what environment was this sample exposed?
Response: The presence of Si in after-adsorption samples is might be due to some contamination. The experiments were conducted in an open lab along with many co-workers who were working on many different projects. Father, this contaminating could also happen during the sample preparation. However, we will be very careful in our future work.